# Digital Toxicology Teleconsultation for Adult Poisoning Cases in Saudi Hospitals: A Nationwide Study

**DOI:** 10.3390/healthcare13050474

**Published:** 2025-02-21

**Authors:** Abdullah A. Alharbi, Mohammed A. Muaddi, Meshary S. Binhotan, Ahmad Y. Alqassim, Ali K. Alsultan, Mohammed S. Arafat, Abdulrahman Aldhabib, Yasser A. Alaska, Eid B. Alwahbi, Ghali Sayedahmed, Mobarak Alharthi, M. Mahmud Khan, Mohammed K. Alabdulaali, Nawfal A. Aljerian

**Affiliations:** 1Family and Community Medicine Department, Faculty of Medicine, Jazan University, Jazan City 45142, Saudi Arabia; aaalharbi@jazanu.edu.sa (A.A.A.); mothman@jazanu.edu.sa (M.A.M.); 2Emergency Medical Services Department, College of Applied Medical Sciences, King Saud bin Abdulaziz University for Health Science, Riyadh 11481, Saudi Arabia; hotanm@ksau-hs.edu.sa (M.S.B.); njerian@moh.gov.sa (N.A.A.); 3King Abdullah International Medical Research Centre, Riyadh 11481, Saudi Arabia; 4Medical Referrals Centre, Ministry of Health, Riyadh 12382, Saudi Arabia; alkalsultan@moh.gov.sa (A.K.A.); arafatms@moh.gov.sa (M.S.A.); aaldhabib@moh.gov.sa (A.A.); ealwahbi@moh.gov.sa (E.B.A.); gsayedahmed@moh.gov.sa (G.S.); mobarakja@moh.gov.sa (M.A.); 5Department of Emergency Medicine, King Saud University, Riyadh 11461, Saudi Arabia; yalaska@ksu.edu.sa; 6Emergency Medicine & Medical Toxicology Department, King Saud Medical City, Riyadh 12746, Saudi Arabia; 7Department of Health Policy and Management, College of Public Health, University of Georgia, Athens, GA 30602, USA; mahmud.khan@uga.edu; 8Ministry of Health, Riyadh 12382, Saudi Arabia; mal-abdul-aali@moh.gov.sa

**Keywords:** toxic epidemiology, digital health, poison control, Saudi Arabia, adult poisoning, healthcare management, surveillance, telemedicine, public health

## Abstract

**Background/Objectives**: Poisoning represents a significant global public health challenge, particularly with its complex manifestations in adult populations. Understanding regional epidemiology through digital health systems is crucial for developing evidence-based prevention and management strategies. This nationwide study analyzes hospital-based toxicology teleconsultation data from the Toxicology Consultation Service-Saudi Medical Appointments and Referrals Center (TCS-SMARC) platform to characterize the epidemiological patterns, clinical features, and outcomes of adult poisoning cases across Saudi regions. **Methods**: We conducted a retrospective cross-sectional analysis of 6427 adult poisoning cases where hospitals sought teleconsultation from the Saudi Toxicology Consultation Service (TCS) from January to December 2023. Descriptive statistics were used to analyze poisoning rates by demographic characteristics, agents responsible for the poisoning, clinical presentations, and management decisions. Population-adjusted rates were calculated using the national census data. Associations between variables were analyzed using cross-tabulations and chi-square tests. **Results**: Young adults aged 18–35 years constituted most cases (58.67%), with the highest population-adjusted rates observed among those aged 18–24 (5.15 per 10,000). Medicine-related poisonings were the most common across all regions (50.04%), followed by bites and stings (15.31%). Regional analysis indicated relatively uniform poisoning rates across Business Units (BUs) (2.02–2.74 per 10,000). Most cases (87.44%) were asymptomatic, with 91.71% exhibiting normal Glasgow Coma Scale scores, although substance abuse cases had higher rate of severe manifestations (24.34%). Significant seasonal variations were observed (*p* < 0.001), with peak incidents occurring in the summer (29.25%). Management decisions primarily involved hospital observation (40.27%) and admission (30.34%), with agent-specific variations in care requirements (*p* < 0.001). **Conclusions**: This comprehensive analysis demonstrates the effectiveness of Saudi Arabia’s digital health infrastructure in capturing and managing nationwide poisoning data. The integrated digital platform enables real-time surveillance, standardized triage, enhanced access to specialized toxicology services, and coordinated management across diverse geographical contexts. Our findings inform evidence-based recommendations for targeted prevention strategies, particularly for young adults and medicine-related poisonings, while establishing a scalable model for digital health-enabled poisoning management.

## 1. Introduction

Poisoning represents a significant global public health challenge, with global estimating approximately 56,000 total poisoning deaths and 2.8 million disability-adjusted life years lost globally in 2021, of which adults account for 48,000 deaths (85.7%) and 2 million disability-adjusted life years (71.4%) [1]. Despite advances in healthcare systems, mortality rates show marked variation, ranging from 0.7% in high-income to 2% in lower-middle income countries, with adult poisoning constituting a substantial burden on healthcare resources and significantly impacting population health outcomes [1,2,3,4,5]. In developed nations like the United States, poisoning cases represented 19.1% (29,748 cases) of total emergency department visits in 2022, highlighting substantial burden it imposes on the healthcare system [6]. The epidemiological patterns vary considerably across nations due to differences in socioeconomics, demographics, healthcare access, and chemical availability, creating complex surveillance challenges. These diverse patterns and varying healthcare capacities necessitate robust surveillance and management systems, yet healthcare providers face significant obstacles in implementing effective monitoring and response strategies.

These challenges are compounded by several critical factors in poisoning surveillance and management. Inconsistent reporting mechanisms across healthcare facilities, varied clinical protocols between countries, and significant delays in data collection collectively hamper effective surveillance and timely response strategies [7]. To address these limitations, healthcare systems worldwide are increasingly implementing digital technologies and integrated platforms. These innovative solutions enhance real-time poisoning surveillance, streamline case management protocols, and improve patient outcomes through standardized documentation and rapid information sharing [8]. The digital transformation of poisoning services has demonstrated promise in enabling rapid detection of emerging trends and facilitating coordinated responses across healthcare networks [9].

In Saudi Arabia, the implementation of digital health solutions in poisoning management aligns with the nation’s Vision 2030 digital transformation initiative [10,11]. The Saudi Medical Appointments and Referrals Center (SMARC) exemplifies this innovative approach, integrating electronic health records with advanced telemedicine platforms to transform healthcare delivery [12,13]. This comprehensive digital ecosystem has significantly enhanced poisoning surveillance and management, enabling real-time monitoring and rapid response to poisoning incidents across the healthcare network [9]. Nevertheless, poisoning remains a significant health challenge, with medicine-related incidents and household chemical exposures being the predominant causes [14,15]. While previous research has largely characterized pediatric poisoning patterns [16,17,18,19], adult poisoning presents complex challenges requiring distinct management approaches [1,20,21].

While digital surveillance systems have enhanced poisoning management across age groups, understanding the distinct characteristics between adult and pediatric poisoning cases remains crucial for optimizing healthcare responses. Adult poisoning cases present unique challenges compared to pediatric cases, characterized by different exposure patterns, intent, and clinical outcomes, with predominantly intentional exposures (75–80%) [21,22], higher prevalence of pharmaceutical and substance abuse [22], increased female susceptibility for self-harm attempts, and mortality rates ranging from 1.2% in China to 4.69% in Ethiopia [22,23]. Unlike pediatric exposures, which are predominantly accidental and occur in home settings, adult cases often involve complex comorbidities [22], intentional exposures, multiple substances, and complex clinical presentations requiring specialized management protocols [24,25]. The variations in toxic agents, exposure circumstances, and treatment approaches between adult and pediatric cases necessitate age-specific surveillance and management strategies. Furthermore, adult poisoning cases often result in higher mortality rates and longer hospital stays, placing a substantial burden on healthcare resources and highlighting the need for targeted intervention strategies [26,27]. This distinction is particularly relevant in the context of digital health systems, where customized workflows and clinical decision support tools must account for these age-specific differences.

The integration of the national Toxicology Consultation Service (TCS) within SMARC in 2020 has transformed poison control management in Saudi Arabia [28]. Operating through SMARC’s digital infrastructure, TCS provides specialized poisoning expertise through real-time consultations and clinical decision support, enabling seamless coordination across healthcare facilities. This integration facilitates nationwide accessibility to poisoning services, particularly crucial given Saudi Arabia’s diverse geographical landscape [9,13]. The SMARC e-referral system has demonstrated significant improvements in healthcare coordination across Saudi Arabia [9,13]. The system has shown particular effectiveness in managing critical and lifesaving cases, with high acceptance rates for intensive care referrals. Its robust data collection and security protocols have enabled comprehensive analysis of referral and poisoning patterns, helping optimize resource allocation and identify areas needing improvement across different regions and specialties [9,13]. Looking ahead, the expected integration of artificial intelligence and machine learning technologies into TCS aligns with Saudi Arabia’s digital transformation vision [13,29].

Leveraging this advanced digital infrastructure, we conducted a comprehensive analysis of adult poisoning cases for which hospitals sought teleconsultation through the TCS and SMARC, hereafter referred to as TCS-SMARC platform. This study aims to assess patterns of toxicology teleconsultations while characterizing the epidemiology of poisoning cases, including demographics, toxic agents, clinical presentations, and management outcomes across Saudi Arabia. The study provides the first nationwide epidemiological analysis through an integrated digital platform, establishing baseline data crucial for evidence-based prevention strategies in hospital-based poisoning care.

## 2. Materials and Methods

This study employed a nationwide retrospective analysis of toxicology teleconsultations to evaluate patterns of toxic exposures and clinical outcomes across Saudi Arabia’s healthcare system. The analysis focused on systematically collected data through the TCS-SMARC platform.

### 2.1. Study Design and Setting

This nationwide retrospective cross-sectional study analyzed adult poisoning cases managed through the Saudi TCS via the SMARC digital platform from 1 January to 31 December 2023. The study included cases where healthcare providers initiated toxicology teleconsultation through the TCS-SMARC platform, representing all cases where clinicians determined specialized toxicology expertise was needed for patient management.

The TCS-SMARC workflow (Figure 1) begins when treating physicians request toxicology consultation through the ‘1937’ hotline. The Specialists in Poison Information (SPI) handle initial triage, leading to one of three decisions: home observation, hospital observation, or escalation to a toxicologist for specialized consultation. When escalated, the toxicologist evaluates the case and provides recommendations based on clinical severity and management needs. These recommendations range from lifesaving referral for critical cases requiring immediate transfer to specialized centers, specific treatment plans for cases manageable at the current facility, home observation for mild cases with adequate home care capability, hospital observation for cases requiring short-term monitoring, or hospital admission for cases needing intensive or extended care. For lifesaving referrals, the Office of Coordination and Eligibility for Treatment (OCET) processes the case through the Unified System of Medical Referrals (USMR). The SPI then follows up until case closure.

### 2.2. Study Population

This nationwide study analyzed 6427 adult (aged ≥ 14 years) hospital-reported poisoning cases managed through the TCS-SMARC digital platform. The study included adult patients (aged 14 years or more), which aligns with Saudi Arabia’s healthcare delivery model [30,31]. The sample size provided adequate statistical power for detecting meaningful differences in poisoning patterns and outcomes, based on previous regional studies reporting similar annual volumes [14,15]. This study encompassed all poisoning teleconsultation cases documented through the digital platform that met the inclusion criteria: teleconsultation requests originated from healthcare facilities, and complete documentation of essential clinical and management variables was available through the digital platform.

### 2.3. Study Variables

We analyzed variables from the SMARC system following standardized national and global health reporting guidelines [32,33,34]. Categorical variables included demographic data (age, sex, nationality) and geographic location (BUs). Toxic agents were classified into medicines, abuse substances, health and nutritional supplements, bites and stings, and household toxics, with an additional category for unclassified agents. Clinical severity assessment incorporated standardized vital sign thresholds: temperature (hypothermia < 36 °C, normothermia 36–37.5 °C, hyperthermia > 37.5 °C), heart rate (bradycardia < 60, normal 60–100, tachycardia > 100 beats/minute), blood pressure (hypertension: systolic ≥ 140 mmHg or diastolic ≥ 90 mmHg sustained > 48 h), oxygen saturation (<95% abnormal), respiratory rate (bradypnea < 12, normal 12–20, tachypnea > 20 breaths/minute), and Glasgow Coma Scale (GCS) scores (normal 13–15, moderate 9–12, severe 8–3) [35,36]. Management decisions followed TCS protocols, including home observation, hospital observation (short-term Emergency Department monitoring for a few hours), hospital admission (formal inpatient admission for more specialized or prolonged medical care), consultation, and lifesaving referral (defined as requiring urgent tertiary care to prevent mortality). Antidote administration was recorded as administered, not required, or not applicable when no antidote existed for the toxic agent [14].

### 2.4. Statistical Analysis

Data were analyzed using STATA version 16.0 (StataCorp LLC, College Station, TX, USA). We used descriptive statistics (frequencies and percentages) to characterize the study population. Population-adjusted rates were calculated using national census data [34]. Statistical significance was set at *p* < 0.05. Associations between toxic agents and demographic characteristics, clinical presentations, and management decisions were analyzed using cross-tabulations with chi-square tests. Subgroup analyses examined clinical presentation patterns and management decisions across different toxic agent categories while controlling for demographic factors.

### 2.5. Ethical Considerations

This study was conducted in accordance with the Declaration of Helsinki and Saudi National Committee of Bioethics (NCBE) guidelines, with approval from the Saudi Ministry of Health Institutional Review Board (Protocol: 23-77-E, approved 20 September 2023). Patient consent requirement was waived due to the retrospective analysis of de-identified data from existing registries. Data were stored on secure servers and accessed only by authorized study team members. Identifiable personal information was removed during extraction for research purposes.

## 3. Results

This nationwide study analyzed poisoning cases reported to the Saudi TCS through the SMARC digital platform. Demographic characteristics and geographic distribution of cases are reported in Table 1. Analysis of hospital-reported poisoning rates showed highest utilization among young adults, with those aged 18–24 years having the highest rate (5.15 per 10,000 population), followed by adolescents aged 14–17 years (4.72 per 10,000 population). While the age group 25–34 years accounted for the largest proportion of consultations (31.01%), their population-adjusted rate was lower (2.61 per 10,000). Gender analysis revealed higher rates among females (3.57 per 10,000) despite their smaller population proportion (35.79%), compared to males (2.05 per 10,000). Regional analysis demonstrated relatively balanced poisoning rates across BUs, ranging from 2.02 to 2.74 per 10,000 population.

Analysis of toxic agent distribution in Table 2 revealed distinct patterns across demographic groups, providing valuable insights into the epidemiology of adult poisoning in Saudi Arabia. Notably, medicine-related poisoning emerged as the predominant toxic agent across all age groups, with particularly high proportions among adolescents aged 14–18 years (58.40%) and young adults aged 18–24 years (61.81%). Furthermore, gender analysis revealed marked differences, with females showing a higher proportion of medicine-related poisonings (66.26%) compared to males (34.30%). Regional analysis demonstrated consistently high rates of medicine-related poisonings across all BUs, with the highest proportions in the Central (53.29%), Eastern (54.55%), and Western (50.31%) regions. However, notable regional variations in secondary agents, such as household toxics in the Eastern region (12.70%) and bites and stings in other regions.

Temporal analysis in Table 3 revealed significant seasonal variations in poisoning incidents (*p* < 0.001), with summer showing the highest frequency (29.25%). Medicine-related poisonings remained the most common toxic agent throughout all seasons. The distribution of secondary toxic agents showed distinct seasonal patterns, with bites and stings emerging as the second most common exposure in spring, summer, and autumn, while winter demonstrated a different pattern with household toxic exposures and substance abuse ranking higher. In contrast, analysis of weekday distribution revealed no statistically significant variations (*p* = 0.068), indicating that poisoning incidents occur relatively uniformly throughout the week.

Analysis of management decisions in Table 4 revealed hospital observation as the predominant approach (40.27%), followed by hospital admission (30.34%). Notably, management patterns varied significantly by toxic agent (*p* < 0.001), with medicine-related poisonings and substance abuse cases frequently requiring hospital admission and observation, while bites/stings, household toxics, and health/nutritional supplements predominantly received hospital observation. Furthermore, the study revealed that antidote administration was necessary for only 11.79% of all cases, with the highest utilization in bites and stings cases (28.46%).

Clinical presentation analysis in Table 5 revealed that the majority of patients (87.44%) remained asymptomatic regardless of toxic agent exposure. Notably, vital sign assessment showed predominantly normal patterns across all toxic agents, with 78.43% maintaining normothermia and 58.57% exhibiting normal heart rates. However, substance abuse cases demonstrated high rates of hyperthermia (5.39%) and tachycardia (37.81%). Furthermore, neurological assessment revealed that while most cases (91.71%) had normal GCS scores, substance abuse and ‘others and unknown’ toxins were associated with higher rates of moderate to severe GCS alterations.

## 4. Discussion

This first nationwide analysis through the TCS-SMARC digital platform provides comprehensive baseline data on adult poisoning cases requiring specialized toxicology teleconsultation across Saudi Arabia’s regions, offering insights into the epidemiological patterns and clinical characteristics of cases requiring specialized toxicology expertise. Our analysis revealed that medicine-related poisonings predominated across all regions, with young adults showing the highest population-adjusted poisoning rates. The digital infrastructure achieved uniform regional coverage despite varying population densities, demonstrating effectiveness in standardizing care protocols and enabling real-time toxicosurveillance across diverse geographical contexts. Unlike previous studies focused on pediatric cases or basic phone consultations [17,19], this comprehensive evaluation provides unprecedented insights into clinical patterns and outcomes of adult poisoning cases. Our findings inform critical healthcare improvements by supporting the implementation of digital prescription monitoring and targeted prevention programs. The data further justifies the development of standardized management protocols and enhanced provider training initiatives. Moreover, these results demonstrate the need to strengthen toxicosurveillance systems and establish substance-specific regulatory frameworks, particularly in regions with elevated risk profiles.

Demographic analysis revealed distinctive service utilization patterns across age groups, with highest engagement among young adults aged 18–24 years, followed by adolescents aged 14–17 years. Our study classified patients aged 14–17 years as adults, reflecting Saudi Arabia’s healthcare delivery model [30,31], though this differs from international systems where pediatric care often extends beyond age 17. This context is important when interpreting our findings and comparing them with international studies. This age distribution aligns with established research showing increased risk-taking behaviors and mental health challenges during these developmental stages [37]. Notably, while the 25–34 age group constituted the largest proportion of consultations, their population-adjusted rate was lower (2.61 per 10,000), likely reflecting both better coping mechanisms and improved health literacy in this age group [34]. Gender analysis showed higher female utilization rates (3.57 vs. 2.05 per 10,000), consistent with established healthcare-seeking behaviors [38,39]. Clinical severity analysis revealed that while most cases were asymptomatic, substance abuse cases demonstrated notably higher rates of severe manifestations. These patterns provide crucial insights for developing targeted prevention strategies within the healthcare system.

Analysis revealed consistent poisoning surveillance coverage across Saudi Arabia’s regions. The Western BU showed slightly higher utilization, followed by Northern and Central BUs. The consistency in toxicosurveillance coverage reflect a successful implementation of the standardized digital infrastructure [9,13]. Medicine-related poisonings predominated across regions, with notably higher prevalence in urban areas, a pattern consistent with other metropolitan healthcare systems [40,41]. The Northern region displayed distinct patterns, with elevated rates of bites and stings and household chemical exposures reflecting its unique environmental and socioeconomic factors [22]. The summer peak aligns with increased outdoor activities and higher temperatures affecting both human behavior and wildlife activity [42,43]. Temporal analysis revealed significant seasonal variations, with peak incidents during summer months [44]. This pattern is likely due to the region’s rural characteristics and increased exposure to wildlife habitats [34]. Bites and stings predominated from spring through autumn, corresponding to the active periods of venomous creatures in warmer months [45] while household toxics and substance abuse showed higher winter prevalence in urban centers, potentially due to increased indoor time and seasonal affective patterns. These geographical and temporal patterns highlight the need for regionally tailored and seasonally adapted prevention strategies, representing an advancement from traditional retrospective surveillance methods [9,13].

The TCS-SMARC digital platform enabled comprehensive analysis of clinical-management correlations, revealing patterns crucial for optimizing poisoning care protocols. Integration of real-time clinical data demonstrated that most cases (87.44%) initially presented as asymptomatic [46,47], which is characteristic of many toxicological exposures due to delayed onset of symptoms and the latency period between exposure and clinical manifestations [48]. The platform’s standardized vital sign monitoring showed predominantly normal patterns, though a high prevalence of hypertension suggested underlying sympathetic activation and anxiety responses [49,50]. Digital documentation of neurological status revealed normal Glasgow Coma Scale (GCS) scores in the majority of cases, with substance abuse presentations showing significantly higher rates of moderate-severe alterations (24.34%) [51], likely due to the direct central nervous system effects of commonly abused substances [52]. Medicine-related poisonings, despite their prevalence, rarely progressed to severe GCS alterations (2.27%), reflecting either low toxicity profiles or successful early interventions enabled by the digital alert system [24,53].

Management decisions, guided by the platform’s integrated clinical decision support tools, aligned closely with presentation severity. As illustrated in Figure 1, the centralized pre-analysis system through TCS enabled standardized triage decisions across facilities, optimizing emergency department utilization patterns. The system facilitated appropriate triage to hospital observation (40.27%) and admission (30.34%) based on agent-specific protocols. Digital tracking revealed targeted intervention patterns: medicine-related and substance abuse cases predominantly required admission [54], likely due to the need for extended monitoring and intervention for these potentially severe exposures, while bites and stings and household toxics typically warranted observation, which could reflect their generally shorter clinical course and response to initial interventions. This systematic approach, while limited to cases where facilities sought TCS consultation, demonstrated the value of centralized toxicology expertise in standardizing care pathways, managing admission flows, and enhancing nationwide access to specialized toxicology services. While antidote administration was tracked quantitatively through the platform, particularly in bites and stings cases, detailed analysis of administration appropriateness was beyond the scope of this study. Real-time monitoring showed minimal cases suitable for home observation, highlighting the complexity of hospital-reported cases. The digital infrastructure’s uniform weekly case distribution enabled optimal resource allocation while identifying opportunities for protocol enhancement [53,55].

The comprehensive digital surveillance through TCS-SMARC has enabled evidence-based insights that inform strategic recommendations for poisoning prevention and management in Saudi Arabia. High hospital-reported poisoning rates among young adults and adolescents, particularly involving medicine-related exposures, underscore the need for targeted interventions through real-time electronic prescription monitoring and age-specific digital health education platforms [56,57], a strategy that has shown success in reducing medication misuse in similar demographic groups internationally [58]. The platform’s success in achieving uniform regional coverage establishes a robust foundation for system-wide enhancements through artificial intelligence-assisted triage protocols and predictive analytics [13]. Region-specific challenges, particularly in the Northern region with its elevated rates of bites and stings and household chemical exposures, necessitate targeted interventions through enhanced environmental monitoring and specialized healthcare provider training [22].

### 4.1. Future Directions

Implementation of these enhancements should follow a systematic, phased approach, progressing from advanced digital surveillance to automated alert mechanisms and predictive analytics [59]. Priority areas for future research include developing machine learning algorithms for real-time risk stratification, integrating genomic data for personalized poisoning management, conducting longitudinal outcome studies, and performing economic analyses of prevention strategies [60]. The integration of the comprehensive national poisoning registry with standardized outcome measures, coupled with enhanced telehealth services, would further strengthen healthcare delivery and surveillance capabilities [61]. This integrated digital health approach provides a scalable model for healthcare systems worldwide facing similar geographic and demographic challenges, particularly in regions with diverse population distributions and varying healthcare access patterns.

### 4.2. Strengths and Limitations

This study has several key strengths that enhance its contribution to toxicoepidemiology research. First, the relatively large sample size and nationwide coverage across Saudi Arabia’s five BUs provide robust statistical power and comprehensive geographical representation. Second, the TCS-SMARC digital platform enabled comprehensive documentation of clinical presentations, vital signs, and management decisions, providing standardized data collection for evidence-based protocol development [9,13]. Third, our focus on adult poisoning addresses a critical knowledge gap, as previous studies predominantly examined pediatric cases [17,19]. Fourth, the inclusion of population-adjusted rates allows for more accurate interpretation of demographic patterns and hospital-reported poisoning service utilization trends. However, the retrospective nature of the study limits our ability to establish direct causality between variables and capture certain clinical details.

## 5. Conclusions

This first nationwide analysis of adult poisoning through Saudi Arabia’s TCS-SMARC digital platform revealed distinct patterns in poisoning cases. Young adults comprised the majority of cases, with medication-related poisonings being most prevalent across all regions. The integrated system achieved uniform coverage and standardized care delivery for reported cases across business units, enabling real-time surveillance of exposure patterns and clinical presentations. Our findings support recommendations for enhanced digital prescription monitoring, targeted prevention programs, and region-specific management protocols. The platform enhanced access to specialized poisoning care, providing a scalable model for healthcare systems with similar geographic challenges while establishing a framework for evidence-based poisoning management.

## Figures and Tables

**Figure 1 healthcare-13-00474-f001:**
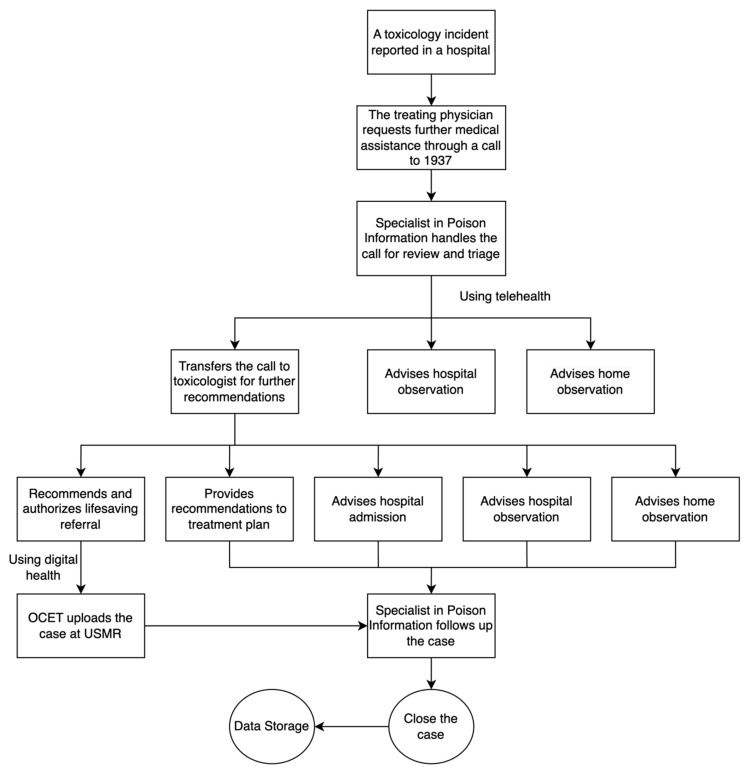
The flow process of the services provided by the Saudi Toxicology Consultation Service (TCS) through the Saudi Medical Appointments and Referrals Center (SMARC) digital platform. Note: Data presented include cases reported to the Toxicology Consultation Service (TCS) through healthcare provider teleconsultation requests and do not represent all poisoning cases in Saudi Arabia. While TCS teleconsultation is available for poisoning cases in Ministry of Health facilities, treating physicians initiate contact based on their assessment of the need for specialized toxicology guidance.

**Table 1 healthcare-13-00474-t001:** Sociodemographic characteristics of poisoning cases and population-based hospital-reported poisoning rates.

Characteristics	Teleconsultation Cases N (%) 6427 (100.00)	Population N (%) 24,775,277 (100%)	Consultation Rate per 10,000 Population
**Age**			
14–17	873 (13.58)	1,849,514 (7.47)	4.72
18–24	1778 (27.66)	3,450,111 (13.93)	5.15
25–34	1993 (31.01)	7,638,547 (30.83)	2.61
35–44	937 (14.58)	6,118,833 (24.70)	1.53
45–54	440 (6.85)	3,213,552 (12.97)	1.37
55–64	237 (3.69)	1,642,871 (6.63)	1.44
65 and more	169 (2.63)	861,849 (3.48)	1.96
**Gender**			
Female	3165 (49.25)	8,868,214 (35.79)	3.57
Male	3262 (50.75)	15,907,063 (64.21)	2.05
**BU**			
Central BU	2246 (34.95)	9,927,927 (40.07)	2.26
Eastern BU	803 (12.49)	3,982,196 (16.07)	2.02
Western BU	2234 (34.76)	8,161,755 (32.94)	2.74
Southern BU	647 (10.07)	2,948,556 (11.90)	2.19
Northern BU	497 (7.73)	1,863,646 (7.52)	2.67

N: frequency; %: percentage; BU: Business Unit. Source: General Authority for Statistics, Saudi Arabia (2023) [34].

**Table 2 healthcare-13-00474-t002:** Distribution of sociodemographic characteristics among toxic agents.

Variable	Total N (%) 6427 (100.00)	Medicines N (%) 3216 (50.04)	Abuse Substances N (%) 612 (9.25)	Health and Nutritional Supplements 258 (4.01)	Bites and Stings N (%) 984 (15.31)	Household Toxics N (%) 680 (10.58)	Others and Unknown N (%) 677 (10.53)	*p*-Value
**Age Groups (Years)**								
14–18	873 (13.58)	510 (58.4)	14 (1.60)	58 (6.64)	118 (13.52)	109 (12.49)	64 (7.33)	0.0000
18–25	1778 (27.66)	1099 (61.81)	126 (7.09)	78 (4.39)	175 (9.84)	158 (8.89)	142 (7.99)	
26–35	1993 (31.01)	948 (47.5)	270 (13.55)	57 (2.86)	290 (14.55)	204 (10.24)	224 (11.24)	
36–45	937 (14.58)	374 (39.91)	126 (13.45)	34 (3.63)	193 (20.60)	112 (11.95)	98 (10.46)	
46–55	440 (6.85)	142 (32.27)	53 (12.05)	14 (3.18)	124 (28.18)	41 (9.32)	66 (15.00)	
56–65	237 (3.69)	90 (37.97)	16 (6.75)	11 (4.64)	46 (19.41)	26 (10.97)	48 (20.25)	
>65	169 (2.63)	53 (31.36)	7 (4.14)	6 (3.55)	38 (22.49)	30 (17.75)	35 (20.71)	
**Sex**								
Female	3165 (49.25)	2097 (66.26)	41 (1.30)	186 (5.88)	270 (8.53)	309 (9.76)	262 (8.28)	0.0000
Male	3262 (50.75)	1119 (34.30)	571 (17.50)	72 (2.21)	714 (21.89)	371 (11.37)	415 (12.72)	
**BU**								
Central	2246 (34.95)	0.0000	227 (10.11)	105 (4.67)	280 (12.47)	197 (8.77)	240 (10.69)	0.0000
Eastern	803 (12.49)	438 (54.55)	53 (6.60)	48 (5.98)	85 (10.59)	102 (12.70)	244 (10.92)	
Western	2234 (34.76)	1124 (50.31)	225 (10.07)	75 (3.36)	370 (16.56)	196 (8.77)	200 (10.64)	
Northern	497 (7.73)	165 (33.20)	34 (6.84)	18 (3.62)	125 (25.15)	107 (21.53)	48 (9.66)	
Southern	647 (10.07)	292 (45.13)	73 (11.28)	12 (1.85)	124 (19.17)	78 (12.06)	68 (10.51)	

N: frequency; %: percentage; BU: Business Unit.

**Table 3 healthcare-13-00474-t003:** Seasonal and weekdays trends of toxic cases.

Variable	Total N (%) 6427 (100.00)	Medicines N (%) 3216 (50.04)	Abuse Substances N (%) 612 (9.25)	Health and Nutritional Supplements 258 (4.01)	Bites and Stings N (%) 984 (15.31)	Household Toxics N (%) 680 (10.58)	Others and Unknown N (%) 677 (10.53)	*p*-Value
**Season**								
Winter	1368 (21.29)	739 (54.02)	177 (12.94)	57 (4.17)	88 (6.43)	178 (13.01)	129 (9.43)	0.0000
Spring	1634 (25.42)	839 (51.35)	179 (10.95)	57 (3.49)	215 (13.16)	158 (9.67)	186 (11.38)	
Summer	1880 (29.25)	833 (44.31)	154 (8.19)	84 (4.47)	414 (22.02)	195 (10.37)	200 (10.64)	
Autumn	1545 (24.04)	805 (52.10)	102 (6.60)	60 (3.88)	267 (17.28)	149 (9.64)	162 (10.49)	
**Weekday**								
Sunday	941 (14.64)	494 (52.50)	78 (8.29)	42 (4.46)	128 (13.60)	110 (11.69)	89 (9.46)	0.0683
Monday	953 (14.83)	486 (51.00)	89 (9.34)	32 (3.36)	136 (14.27)	93 (9.76)	117 (12.28)	
Tuesday	882 (13.72)	468 (53.06)	90 (10.20)	32 (3.63)	120 (13.61)	88 (9.98)	84 (9.52)	
Wednesday	835 (12.99)	392 (46.95)	81 (9.70)	43 (5.15)	139 (16.65)	91 (10.90)	89 (10.66)	
Thursday	910 (14.16)	457 (50.22)	93 (10.22)	49 (5.38)	133 (14.62)	94 (10.33)	84 (9.23)	
Friday	923 (14.36)	437 (47.35)	96 (10.40)	28 (3.03)	156 (16.90)	97 (10.51)	109 (11.81)	
Saturday	983 (15.29)	482 (49.03)	85 (8.65)	32 (3.26)	172 (17.50)	107 (10.89)	105 (10.68)	983 (15.29)

N: frequency; %: percentage.

**Table 4 healthcare-13-00474-t004:** Correlation between toxic agents, medical management approach, and antidote use.

Variable	Total N (%) 6427 (100.00)	Medicines N (%) 3216 (50.04)	Abuse Substances N (%) 612 (9.25)	Health and Nutritional Supplements 258 (4.01)	Bites and Stings N (%) 984 (15.31)	Household Toxics N (%) 680 (10.58)	Others and Unknown N (%) 677 (10.53)	*p*-Value
**Decision by Agent**								
Home observation	273 (4.25)	129 (4.01)	9 (1.47)	33 (12.79)	19 (1.93)	53 (7.79)	30 (4.43)	0.0000
Hospital observation	2588 (40.27)	1119 (34.79)	212 (34.64)	106 (41.09)	540 (54.88)	347 (51.03)	264 (39.00)	
Hospital admission	1950 (30.34)	1191 (37.03)	219 (35.78)	58 (22.48)	190 (19.31)	106 (15.59)	186 (27.47)	
Lifesaving referral	348 (5.41)	158 (4.91)	41 (6.70)	7 (2.71)	57 (5.79)	49 (7.21)	36 (5.32)	
Consultation	1268 (19.73)	619 (19.25)	131 (21.41)	54 (20.93)	178 (18.09)	125 (18.38)	161 (23.78)	
**Antidote**								
No	2134 (33.20)	1088 (33.83)	184 (30.07)	91 (35.27)	364 (36.99)	215 (31.62)	192 (28.36)	0.0000
Not Applicable	3535 (55.00)	1809 (56.25)	364 (59.48)	152 (58.91)	340 (34.55)	433 (63.68)	437 (64.55)	
Yes	758 (11.79)	319 (9.92)	64 (10.46)	15 (5.81)	280 (28.46)	32 (4.71)	48 (7.09)	

N: frequency; %: percentage.

**Table 5 healthcare-13-00474-t005:** Clinical presentations of patients experiencing poisoning incidents.

Variable	Total N (%) 6427 (100.00)	Medicines N (%) 3216 (50.04)	Abuse Substances N (%) 612 (9.25)	Health and Nutritional Supplements 258 (4.01)	Bites and Stings N (%) 984 (15.31)	Household Toxics N (%) 680 (10.58)	Others and Unknown N (%) 677 (10.53)	*p*-Value
**Symptoms**								
Asymptomatic	5620 (87.44)	2771 (86.16)	541 (88.40)	232 (89.92)	874 (88.82)	603 (88.68)	599 (88.48)	0.0736
Symptomatic	807 (12.56)	445 (13.84)	71 (11.60)	26 (10.08)	110 (11.18)	77 (11.32)	78 (11.52)	
**Temperature**								
Hypothermia	1142 (17.77)	547 (17.01)	116 (18.95)	48 (18.60)	169 (17.17)	123 (18.09)	139 (20.53)	0.0342
Normothermia	5041 (78.43)	2560 (79.60)	463 (75.65)	200 (77.52)	772 (78.46)	541 (79.56)	505 (74.59)	
Hyperthermia	244 (3.80)	109 (3.39)	33 (5.39)	10 (3.88)	43 (4.37)	16 (2.35)	33 (4.87)	
**Pulse rate**								
Bradycardia	251 (3.92)	102 (3.19)	19 (3.11)	7 (2.76)	63 (6.40)	19 (2.81)	41 (6.06)	0.0000
Normal rate	3747 (58.57)	1738 (54.40)	361 (59.08)	160 (62.99)	718 (72.97)	378 (55.92)	392 (57.90)	
Tachycardia	2399 (37.50)	1355 (42.41)	231 (37.81)	87 (34.25)	203 (20.63)	279 (41.27)	244 (36.04)	
**Hypertension**								
No	1102 (17.15)	520 (16.17)	102 (16.67)	39 (15.12)	161 (16.36)	136 (20.00)	144 (21.27)	0.0086
Yes	5325 (82.85)	2696 (83.83)	510 (83.33)	219 (84.88)	823 (83.64)	544 (80.00)	533 (78.73)	
**Glasgow coma scale**								
Normal	5894 (91.71)	2998 (93.22)	463 (75.65)	253 (98.06)	979 (99.49)	655 (96.32)	546 (80.65)	0.0000
Moderate	311 (4.84)	145 (4.51)	75 (12.25)	5 (1.94)	3 (0.30)	14 (2.06)	69 (10.19)	
Severe	222 (3.45)	73 (2.27)	74 (12.09)	0 (0.00)	2 (0.20)	11 (1.62)	62 (9.16)	
**Blood oxygen saturation**								
abnormal	477 (7.42)	175 (5.44)	86 (14.05)	8 (3.10)	66 (6.71)	68 (10.00)	74 (10.93)	0.0000
Normal	5950 (92.58)	3041 (94.56)	526 (85.95)	250 (96.90)	918 (93.29)	612 (90.00)	603 (89.07)	
**Respiratory rate**								
Bradypnea	313 (4.87)	154 (4.79)	28 (4.58)	12 (4.65)	52 (5.28)	27 (3.97)	40 (5.91)	0.0000
Normal	4363 (67.89)	2253 (70.06)	385 (62.91)	169 (65.50)	710 (72.15)	384 (56.47)	462 (68.24)	
Tachypnea	1751 (27.24)	809 (25.16)	199 (32.52)	77 (29.84)	222 (22.56)	269 (39.56)	175 (25.85)	

N: frequency; %: percentage.

## Data Availability

The data presented in this study are available on request from the corresponding author because the data are part of an ongoing mega project. All raw data supporting the conclusions of this article will be made available upon review of the request. Any inquiries about data access should be directed to the corresponding author via institutional email.

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
