# Peer review of "Digital Toxicology Teleconsultation for Adult Poisoning Cases in Saudi Hospitals: A Nationwide Study"

_healthcare, 2025, doi:10.3390/healthcare13050474_

Round 1
Reviewer 1 Report
Comments and Suggestions for Authors
While the study effectively utilizes Saudi Arabia’s digital health infrastructure for poison control analysis, addressing concerns regarding novelty, methodological depth, and the use of more advanced analytical models would significantly enhance its scientific contribution and practical relevance.

Author Response
Response to First Reviewer
Comment 1: The study leverages Saudi Arabia's integrated digital health infrastructure to analyze adult poisoning cases reported to hospitals nationwide. It aims to examine demographic characteristics, poisoning agents, clinical presentations, and management outcomes while assessing the effectiveness of a digitally enabled poison control system.
Although the study presents a comprehensive data analysis, its novel contribution is unclear, as it primarily relies on descriptive statistics without introducing new analytical approaches or methodologies.
Response 1:
Thank you for this insightful comment. Our study utilizes descriptive statistics, but its novelty lies in providing the first comprehensive nationwide epidemiological analysis of adult poisoning cases managed through digital toxicology teleconsultation in Saudi Arabia. We have refined our manuscript as follows:
- Abstract:
" This nationwide study analyzes hospital-based toxicology teleconsultation data from the Toxicology Consultation Service- Saudi Medical Appointments and Referrals Center (TCS-SMARC) platform to characterize the epidemiological patterns, clinical features, and outcomes of adult poisoning cases across Saudi regions. "
- Introduction (final paragraph):
“This study aims to assess patterns of toxicology teleconsultations while characterizing the epidemiology of poisoning cases, including demographics, toxic agents, clinical presentations, and management outcomes across Saudi Arabia. The study provides the first nationwide epidemiological analysis through an integrated digital platform, establishing crucial baseline data for evidence-based prevention strategies in hospital-based poisoning care.”
- Discussion:
“This first nationwide analysis through the TCS-SMARC digital platform provides comprehensive baseline data on adult poisoning cases requiring specialized toxicology teleconsultation across Saudi Arabia's regions, offering insights into the epidemiological patterns and clinical characteristics of cases requiring specialized toxicology expertise.”
Comment 2: The study could benefit from incorporating more advanced data analysis techniques, such as data mining algorithms and machine learning models, to uncover deeper patterns and novel insights from the dataset. While descriptive statistics provide an overview of poisoning cases, the absence of predictive or inferential analysis limits the study’s impact. The inclusion of trend analysis, clustering techniques, or predictive modeling could significantly improve its applicability in poison control management.
Response 2:
Thank you for bringing attention to our methodology. Our analytical approach aligns with our primary objective: evaluating regional patterns of toxicology teleconsultations while characterizing the epidemiology of poisoning cases across Saudi Arabia through the TCS-SMARC digital platform.
Our outcome variables (toxic agent groups) are categorical, and our data represents hospital-based toxicology teleconsultation encounters. In this context, univariate and bivariate analyses effectively serve our research aims. The study's strength is enhanced by our large nationwide sample and comprehensive coverage through the integrated digital platform, ensuring robust statistical power and generalizability of findings.
While we acknowledge that future research would benefit from more sophisticated approaches (machine learning, multilevel modeling) to investigate predictors of patient outcomes, our current methodology is appropriately matched to our objective of establishing baseline epidemiological patterns of poisoning cases requiring toxicology teleconsultation. We believed that our current approach provides both scientific rigor and practical utility while maintaining analytical validity.
The study's novelty lies in providing the first comprehensive nationwide epidemiological analysis of adult poisoning cases managed through the TCS-SMARC digital platform in Saudi Arabia. This baseline data establishes crucial benchmarks for toxicology teleconsultation services while laying the groundwork for more advanced analytical approaches in future research and evidence-based prevention strategies.
Reviewer 2 Report
Comments and Suggestions for Authors
The study is very interesting and well developed. The rationale flows well.

Author Response
Response to Second Reviewer
Comment 1: Title: Digital Surveillance of Adult Poisoning Cases in Saudi 2 Hospitals: A Nationwide Study
Poisoning represents a significant global public health challenge, particularly with its complex manifestations in adult populations. The AUTHORS studied Saudi Arabia's integrated digital health infrastructure to analyze adult poisoning cases reported to hospitals nationwide.
Retrospective cross-sectional analysis of 6427 cases of poisoning was carried out where hospitals sought teleconsultation from the Saudi Toxicology Consultation Service (TCS) from January to December 2023.
Descriptive statistics were used to analyze poisoning rates by demographic characteristics, agents responsible for the poisoning, clinical presentations, and management decisions. Population-adjusted rates were calculated using the updated national census data.
AUTHORS’ results of this study show that young adults aged 18-35 years constituted most cases (58.67%), medicine-related poisonings were the most common across all 35 regions (50.04%), followed by bites and stings (15.31%). Most cases (87.44%) were asymptomatic, with 91.71% exhibiting normal Glasgow Coma Scale scores, although substance abuse cases had higher rate of severe manifestations (24.34%). Management decisions primarily involved hospital observation (40.27 %) and admission (30.34%), with agent-specific variations in care requirements (p<0.001). The study is very interesting and well developed. The rationale flows well.
Response 1: We appreciate your positive evaluation of our manuscript.
Comment 2: I only have one major doubt that needs clarification regarding the database used. I believe that a Poison Control Center should refer to a toxicological database together with a database of antidotes of proven efficacy. In the study it seems to me that this explicit reference is missing.
Response 2:
Thank you for raising this important point about toxicological and antidote databases. Our study utilized secondary data collected by healthcare providers and reported to the toxicology center. While providers documented whether patients received antidotes, a limitation of this secondary data is that specific details about the types and kinds of antidotes administered were not consistently captured in the reporting system.
We acknowledge that a comprehensive Poison Control Center database would ideally include detailed information about both toxicological agents and their corresponding antidotes and would enhance the clinical utility of the findings. We have noted this as a limitation in our study.
Despite this limitation, we believed that our dataset provides valuable epidemiological insights into toxic exposures across Saudi Arabia, serving as a foundation for future enhanced surveillance systems that could incorporate more detailed antidote tracking.
Comment 3: In the introduction develop the bibliographic references (see for example on line 59 “burden on 59 healthcare resources and significantly impacting population health outcomes [1-4].”
Response 3:
IN RED, we have updated the bibliographic references throughout the manuscript.
Comment 4: In the introduction lacks an effective aim “Our findings not only inform evidence-based…”
Response 4:
We have revised the introduction to provide a clear, structured presentation of the study objective. The revised text now reads:
This study aims to assess patterns of toxicology teleconsultations while characterizing the epidemiology of poisoning cases, including demographics, toxic agents, clinical presentations, and management outcomes across Saudi Arabia. The study provides the first nationwide epidemiological analysis through an integrated digital platform, establishing crucial baseline data for evidence-based prevention strategies in hospital-based poisoning care.
Comment 5: Figure 1 must be described in detail in the body of the manuscript.
Response 5:
We have added a detailed description of Figure 1 in the manuscript as follows:
The TCS-SMARC workflow (Figure 1) begins when treating physicians request toxicology consultation through the '1937' hotline. The Specialists in Poison Information (SPI) handle initial triage, leading to one of three decisions: home observation, hospital observation, or escalation to a toxicologist for specialized consultation. When escalated, the toxicologist evaluates the case and provides recommendations based on clinical severity and management needs. These recommendations range from lifesaving referral for critical cases requiring immediate transfer to specialized centers, specific treatment plans for cases manageable at the current facility, home observation for mild cases with adequate home care capability, hospital observation for cases requiring short-term monitoring, or hospital admission for cases needing intensive or extended care. For lifesaving referrals, the Office of Coordination and Eligibility for Treatment (OCET) processes the case through the Unified System of Medical Referrals (USMR). The SPI then follows up until case closure.
Comment 6: A small summary of the study design before immediately entering the subsections dedicated to the design would.
Response 6:
We added a summary before Section 2.1 as follows:
This study employed a nationwide retrospective analysis of toxicology tele-consultations to evaluate patterns of toxic exposures and clinical outcomes across Saudi Arabia's healthcare system. The analysis focused on systematically collected data through the TCS-SMARC platform.
Reviewer 3 Report
Comments and Suggestions for Authors
The article analyses the surveillance for adult poisoning (> 14 years) in Saudi Arabia in the year 2023 through a centralised and computerised national surveillance system.
It is relevant to the purpose of the Journal and valid for comparison and significance, also given the nation-wide sample.
I would have a few minor observations to implement the work, which is overall worthy and methodologically sound.
First point: purpose: I consider the purpose of the surveillance to be correct, but to verify, seeing the authors' own limitations, with one year, the effectiveness of the system seems to me not entirely correct. Perhaps it would be worth limiting the aim to the study of the poisoning phenomenon. The considerations on the efficacy of the system to be left for discussion further .
Abstract: revise aim as above and revise conclusions as above as overstated and additional to methods and results.
references 15-18 : please select the more update and consistent ones, e.g. remove 15 and 17 in preference to the systematic review. same thing references 26-28, select the work which best synthesises also to avoid excessive self-citing.
line 119 to 124 not relevant with purpose and results and therefore not useful for defining the background since surveillance does not today as described foresee a role for IA. Reference 12 not relevant therefore and prefer removal.
figure 1 and text: please clarify whether reporting of poisoning is mandatory, to support the strength of the data derived from this surveillance system.
setting: please clarify what is meant by hospital observation to better understand the difference with hospitalisation.
suggestion: Figure 1, with the inclusion of some results on hospital admission, observation, home observation referral ... could be very useful as a graphical abstract as well as very immediate as data and catchy for the reader and reproduction in other contexts.
references 33s move after the full stop in line 194
results: detail a bit if possible category others or unknown toxic
discussion: emphasise the fact that adults is from 14 years and discuss results in that 14-17 range considering how many countries would place this portion in paediatric.
line 336 following. I would discuss the figure further, also with respect to the appropriate limitations mentioned, as it emphasises the importance of a centralised pre-analysis system for defining the indication for access to an emergency room, thus better managing admission flows.
line 314-344 review as the antidote data is only quantitative, there is no result on the indication and thus on the correctness of the administration.
line 356-357 not applicable and ref. 59 neither.
thanks
Author Response
Response to Third Reviewer
The article analyses the surveillance for adult poisoning (> 14 years) in Saudi Arabia in the year 2023 through a centralised and computerised national surveillance system.
It is relevant to the purpose of the Journal and valid for comparison and significance, also given the nation-wide sample.
Comment 1: I would have a few minor observations to implement the work, which is overall worthy and methodologically sound.
First point: purpose: I consider the purpose of the surveillance to be correct, but to verify, seeing the authors' own limitations, with one year, the effectiveness of the system seems to me not entirely correct. Perhaps it would be worth limiting the aim to the study of the poisoning phenomenon. The considerations on the efficacy of the system to be left for discussion further .
Abstract: revise aim as above and revise conclusions as above as overstated and additional to methods and results.
Response 1:
We appreciate your thoughtful observation. We have rephrased our aim statements to focus more precisely on the poisoning phenomenon for cases documented through teleconsultation.
We have modified both the abstract and main text to reflect this more focused scope. The conclusions have been adjusted to align more closely with the modifications.
The revised aim statement is refined in our manuscript as follows:
- Abstract:
"This nationwide study analyzes hospital-based toxicology teleconsultation data from the Toxicology Consultation Service- Saudi Medical Appointments and Referrals Center (TCS-SMARC) platform to characterize the epidemiological patterns, clinical features, and outcomes of adult poisoning cases across Saudi regions."
- Introduction (final paragraph):
“This study aims to assess patterns of toxicology teleconsultations while characterizing the epidemiology of poisoning cases, including demographics, toxic agents, clinical presentations, and management outcomes across Saudi Arabia. The study provides the first nationwide epidemiological analysis through an integrated digital platform, establishing crucial baseline data for evidence-based prevention strategies in hospital-based poisoning care.”
- Discussion:
“This first nationwide analysis through the TCS-SMARC digital platform provides comprehensive baseline data on adult poisoning cases requiring specialized toxicology teleconsultation across Saudi Arabia's regions, offering insights into the epidemiological patterns and clinical characteristics of cases requiring specialized toxicology expertise.”
Comment 2: references 15-18 : please select the more update and consistent ones, e.g. remove 15 and 17 in preference to the systematic review. same thing references 26-28, select the work which best synthesises also to avoid excessive self-citing.
Response 2:
We have carefully reviewed and refined our citations, keeping the most comprehensive systematic review while removing citations 15 and 17. We also chose the most representative work from references 26-28, which helped reduce self-citation and provided a more focused literature foundation.
Comment 3: line 119 to 124 not relevant with purpose and results and therefore not useful for defining the background since surveillance does not today as described foresee a role for IA. Reference 12 not relevant therefore and prefer removal.
Response 3:
Thank you for this valuable observation. We have removed lines 119-124 and reference 12
Comment 4: figure 1 and text: please clarify whether reporting of poisoning is mandatory, to support the strength of the data derived from this surveillance system.
Response 4:
Thank you for this important clarification request. We have added explanatory text to both the methods section and figure legend to clearly indicate that our data represents cases where healthcare providers actively sought toxicology consultation through the TCS-SMARC platform, rather than all poisoning cases in Saudi Arabia. The reporting is not mandatory - clinicians engage with the teleconsultation service based on their assessment of when specialized toxicology expertise is needed for patient management.
The added text is:
- Methods (Section 2.1):
"The study included cases where healthcare providers-initiated toxicology teleconsultation through the SMARC-TCS platform, representing all cases where clinicians determined specialized toxicology expertise was needed for patient management."
- Figure 1 Legend:
"Data presented include cases reported to the Toxicology Consultation Service (TCS) through healthcare provider teleconsultation requests and do not represent all poisoning cases in Saudi Arabia. While TCS teleconsultation is available for poisoning cases in Ministry of Health facilities, treating physicians initiate contact based on their assessment of the need for specialized toxicology guidance."
Comment 5: setting: please clarify what is meant by hospital observation to better understand the difference with hospitalisation.
Response 5:
Thank you for your valuable comment regarding the distinction between hospital observation and hospitalization. In our study context, these terms reflect different levels of care based on clinical severity and monitoring requirements:
- Hospital observation refers to cases where patients are monitored in the Emergency Department for few hours, allowing healthcare providers to ensure clinical stability before discharge. This decision is made based on the type of toxic exposure, presenting symptoms, and standard clinical protocols.
- Hospitalization, in contrast, involves formal admission to an inpatient ward for more intensive or prolonged treatment. This occurs in cases with severe clinical presentations or those requiring extended medical intervention.
Added text to Methods (Section 2.3.):
Management decisions followed TCS protocols, including home observation, hospital observation (short-term Emergency Department monitoring for a few hours), hospital admission (formal inpatient admission for more specialized or prolonged medical care),
Comment 6: suggestion: Figure 1, with the inclusion of some results on hospital admission, observation, home observation referral ... could be very useful as a graphical abstract as well as very immediate as data and catchy for the reader and reproduction in other contexts.
Response 6:
We appreciate this thoughtful suggestion. While Figure 1 serves as a valuable workflow template for future studies, we agree that a graphical abstract would enhance visibility. We have provided a graphical abstract that highlights the key disposition outcomes (hospital admission, observation, home observation, referral) and main findings.
Comment 7: references 33s move after the full stop in line 194
Response 7:
Thank you for the observation. Addressed per comment.
Comment 8: results: detail a bit if possible category others or unknown toxic
Response 8:
Thank you for highlighting this point about the 'unknown/other' toxic category. As our study utilized national secondary data, one inherent limitation was the classification of certain toxic agents as 'unknown' or 'other.' While this is a common limitation in secondary data analysis, it's important to note that these cases represented only approximately 10% of our total sample. The remaining 90% of toxic agents were well-classified according to scientific categories.
This limitation, while acknowledged, does not significantly impact our study's overall validity and strength. We believe that our dataset's robust sample size and comprehensive national coverage, encompassing toxic exposure visits across all regions of Saudi Arabia, provides strong statistical power and representativeness. The well-classified majority of cases allows for meaningful epidemiological insights into toxic exposure patterns across the kingdom.
For transparency in reporting, we maintained these categories as they appeared in the original data, adhering to scientific rigor in our methodology rather than making assumptions about unspecified agents.
Comment 9: discussion: emphasise the fact that adults is from 14 years and discuss results in that 14-17 range considering how many countries would place this portion in paediatric.
Response 9:
In our study, we classified patients aged 14-17 years as adults [1,2], which reflects the standard healthcare delivery model in Saudi Arabia's health system. This classification is operationalized across Saudi healthcare facilities, where patients in this age group are routinely managed in adult care settings, including adult inpatient wards and outpatient departments, rather than pediatric services.
While this age stratification differs from some international healthcare systems that extend pediatric care through age 17, our classification aligns with the existing healthcare delivery structure in Saudi Arabia [1,2]. This context is essential for interpreting our findings and understanding the service delivery patterns observed in our study.
References:
1- Najjar, Shahenaz, et al. "Determinants of adolescents’ perceptions on access to healthcare services in the Kingdom of Saudi Arabia: Jeeluna national survey findings." BMJ open 11.10 (2021): e035315.
2- Alnasser, Yossef S., et al. "Practice of general pediatrics in Saudi Arabia: current status, challenges, and opportunities." BMC pediatrics 22.1 (2022): 621.
- Methods (Section 2.)
“The study included adult patients (aged 14 years or more), which aligns with Saudi Arabia's healthcare delivery modell [30,31].”
- Discussion:
“Our study classified patients aged 14-17 years as adults, reflecting Saudi Arabia's healthcare delivery model [30,31], though this differs from international systems where pediatric care often extends through age 17. This context is important when interpreting our findings and comparing them with international studies.”
Comment 10: line 336 following. I would discuss the figure further, also with respect to the appropriate limitations mentioned, as it emphasises the importance of a centralised pre-analysis system for defining the indication for access to an emergency room, thus better managing admission flows.
Response 10
Thank you for this important observation about discussing Figure 1 in greater detail, particularly regarding its role in demonstrating centralized pre-analysis capabilities. We have made two key additions to strengthen this aspect of the manuscript:
- We added the following text early in the paragraph:
"As illustrated in Figure 1, the centralized pre-analysis system through TCS enabled standardized triage decisions across facilities, optimizing emergency department utilization patterns”
- We also added this contextual statement:
"This systematic approach, while limited to cases where facilities sought TCS consultation, demonstrated the value of centralized toxicology expertise in standardizing care pathways and managing admission flows, and enhanced nationwide access to specialized toxicology services."
Comment 11: line 314-344 review as the antidote data is only quantitative, there is no result on the indication and thus on the correctness of the administration.
Response 11:
Thank you for your valuable observation. We have modified the text to be more precise about the scope of our antidote-related findings.
Specifically, we rephrased the text to read:
"While antidote administration was tracked quantitatively through the platform, particularly in bites and stings cases, detailed analysis of administration appropriateness was beyond the scope of this study."
Comment 12: line 356-357 not applicable and ref. 59 neither.
Response 12:
We have removed the statement in line 356-357 and its corresponding reference.